

# Expression and drought functional analysis of one circRNA PecircCDPK from moso bamboo (*Phyllostachys edulis*)

Yiqian Li, Wen Xia, Ying Li and Xueping Li

International Center for Bamboo and Rattan, Key Laboratory of National Forestry and Grassland Administration/Beijing for Bamboo & Rattan Science and Technology, Beijing, China

## ABSTRACT

Drought stress can affect the growth of bamboo. Circle RNAs (CircRNAs) have been found to play a role in drought stress in plants, but their function in moso bamboo is not well understood. In previous studies, we observed that under drought stress, the expression of some circRNAs were altered and predicted to be involved in calcium-dependent protein kinase phosphorylation, as indicated by KEGG enrichment analysis. In this study, we cloned a circRNA called *PecircCDPK* in moso bamboo that is responsive to drought stress. To further investigate its function, we constructed an overexpression vector using flanking intron sequences supplemented by reverse complementary sequences. When this vector was transferred to *Arabidopsis* plants, we observed that the roots of the transgenic lines were more developed, the water loss rate decreased, the stomata became smaller, and the activity of antioxidant enzymes increased under drought stress. These findings suggest that overexpression of *PecircCDPK* can enhance the drought resistance of *Arabidopsis thaliana*, providing valuable insights for the breeding of moso bamboo with improved resistance to drought.

## INTRODUCTION

China possesses approximately one-third of the world's bamboo resources, with moso bamboo being the predominant and unique species. Moso bamboo is widely distributed and holds significant practical value as it can be utilized to produce a diverse range of products (*Tang, 2019*). As the social economy progresses, the demand for moso bamboo products has expanded. However, in recent years, the temperature rise caused by global warming has led to more frequent extreme heat events and long-term droughts, affecting the yield and quantity of bamboo. The severe lack of water resources during the bamboo shoot period can result in slow bamboo shoot emergence, high rate of shoot recession, and low rate of bamboo maturation, which will directly affect the economic benefits of bamboo plants. The dry weather during the bamboo growing period can also cause burns, loss of green and white on bamboo leaves. Consequently, drought stress has emerged as a crucial constraint in enhancing crop yield and quality.

Corresponding authors
Ying Li, liying@icbr.ac.cn
Xueping Li, lxp@icbr.ac.cn

Soil drought and atmospheric drought are currently the most common types. These two types of drought often occur simultaneously during plant growth and development. When drought occurs, plants experience a shortage of water due to the water consumption exceeding absorption (*Hsiao, 1973*). Severe water scarcity exerts drought pressure on plants, reducing the swelling pressure of plant cells, inhibiting cell division, and impeding the normal extension and growth of plant leaves. This leads to symptoms such as withering and curling. Furthermore, numerous studies have investigated the molecular regulatory mechanisms of abiotic stress responses (*Tang & Luan, 2017*; *Hivrale et al., 2016*; *Huang et al., 2017*). One such protein, calcium-dependent protein kinase, possesses dual functions as both a responder and sensor. It can directly sense and respond to calcium ion signals, converting them into downstream protein phosphorylation pathways (*Poovaiah et al., 2013*). This protein plays a crucial role in stomatal movement, root development, and hormone signal transduction (*Wu et al., 2020*). Therefore, conducting research on the drought resistance of moso bamboo is essential.

CircRNAs are a recently discovered type of endogenous non-coding RNA (ncRNA) (*Sanger et al., 1976*). They have a unique closed-loop structure formed by 5′–3′ ligation during splicing, which makes them resistant to degradation by nucleic acid exonucleases. For a long time, circRNAs were considered as splicing errors and were not functional. However, with the advancement of high-throughput sequencing technology and bioinformatics, their important roles in biology are gradually being revealed. In 2011, Danan developed a method to identify circRNAs at the genomic level using RNA-Seq data (*Danan et al., 2011*). Subsequently, *Memczak et al. (2013)* analyzed RNA-Seq data and identified thousands of stable and functional circRNAs in humans, mice, and nematodes. Not only have circRNAs been extensively studied in humans and animals, but they have also been identified in various plants such as *Arabidopsis thaliana*, maize (*Zhang et al., 2019*), and rice (*Lu et al., 2015*). Interestingly, some circRNAs exhibit stage-specific expression during tissue development. For example, *Arabidopsis* plants overexpressing *Vv-circATS1*, a circRNA derived from glycerol-3-P acyltransferase, showed improved cold tolerance compared to those overexpressing the linear RNA sequence of *Vv-circATS1* (*Gao et al., 2019*). Currently, 895 circRNAs have been identified in moso bamboo shoots (*Wang et al., 2019*), and 4931 circRNAs have been found in response to drought stress in moso bamboo (*Li et al., 2022*). This suggests that circRNAs play a role in lignin synthesis and response to drought stress in moso bamboo. Among these circRNAs, *PecircCDPK* has been predicted to have calcium-dependent protein kinase activity and participate in drought stress response, as indicated by KEGG enrichment analysis (*Li et al., 2022*). Therefore, based on the aforementioned speculation, we believe that the parental genes of *PecircCDPK* are involved in drought stress responses, and *PecircCDPK* itself is also involved in regulating stomata, causing changes in plant phenotypes, and other responses to drought stress.

In our study, we cloned *PecircCDPK* and construct an overexpression vector by using the principle of flanking intron sequences supplemented by reverse complementary sequences (*Gao et al., 2019*; *Li et al., 2022*). Phenotypic differences in physiological phenotypic were observed and analyzed between transgenic and wild-type plants under drought conditions.

Our study would provide a new insight into the drought stress tolerance in moso bamboo and lay a foundation for the functional study of circRNAs in bamboo plants in the future.

## MATERIALS AND METHODS

### Plant materials and treatments

Moso bamboo seeds were collected from Guangxi Zhuang Autonomous Region, China and planted in a plastic basin with a diameter of approximately 10 cm, filled with humus soil. The seeds were then incubated in a constant temperature and light incubator, with a day and night temperature of 25/18 °C and a photoperiod of light/dark 16/8 h. The culture lasted for approximately 3 months. To simulate drought stress, the seedlings of moso bamboo were treated with PEG 6000. After 0 h, 6 h, 12 h, 24 h, and 48 h (P1, P2, P3, P4, P5) of treatment, the leaves from the same part were collected.

The wild-type *Arabidopsis thaliana* Columbia-0 was used in this study. First, *Arabidopsis thaliana* seeds were sterilized and sown in 1/2 MS medium. Subsequently, the seeds were vernalized in a refrigerator at 4 °C for 2 days. After the vernalization process, the seeds were transferred to petri dishes and placed in a growth room for a period of 7–10 days. Following this, the *Arabidopsis thaliana* seedlings were transplanted into plastic pots filled with humus soil. The pots were then placed in an incubator set at a temperature of 23 °C and a light/dark photoperiod of 16/8 h. Each sample was subjected to three biological replicates and frozen in liquid nitrogen and stored at −80 °C. All statistical analyses were conducted using SPSS software.

### RNA extraction and cDNA synthesis

Total RNA and DNA was extracted from each sample using the RNAprep Pure Plant Kit (Tiangen, Beijing, China) and FastPure Plant DNA Isolation Mini Kit (Vazyme, Nanjing, China) following the manufacturer's protocol. The purity, integrity and concentration of total RNA and DNA were examined using NanoDrop2000 (NanoDrop Technologies, Wilmington, DE, USA) and gel electrophoresis. A Reverse transcription using PrimeSTAR GXL DNA polymerase (R050A, Takara, Shiga, Japan).

### Plasmid construction and genetic transformation

*Hic_scaffold_3:83696771|83697493* was previously validated and predicted to function as a calcium-dependent protein kinase in our previous work (*Li et al., 2022*). In this study, we have named it *PecircCDPK*. The *PecircCDPK* overexpression vector was constructed by incorporating flanking intron sequences supplemented with reverse complementary sequences into the *PCAMBIAsuper1300-GFP* vector (*Gao et al., 2019*).

The principle of using flanking intron sequences supplemented by reverse complementary sequences is as follows: Firstly, we used the DNA sequence of *PH02Gene31251* as a template and selected a 437 bp intron fragment from it to act as the upstream cyclization sequence. Next, we added *XbaI* and *KpnI* sites at both ends of the upstream cyclization sequence. This modified sequence was then connected to the T Easy vector, transformed into Trans5$\alpha$, and finally connected with the *PCAMBIAsuper1300-GFP* vector. The PCR amplification reaction for the upstream cyclization sequence involved the

following steps: 94 °C for 5 min; 94 °C for 30 s, 55 °C for 30 s, 72 °C for 26 s, repeated for 30 cycles; and storage at 4 °C.

The downstream cyclization sequence is the reverse complementary sequence of the upstream cyclization sequence. *BsrGI* and *EcoRI* sites were added at both ends of the downstream cyclization sequence. The downstream cyclization sequence was then connected to the T Easy vector and transformed into *Trans5 α*. It is also connected with the upstream cyclization sequence-*PCAMBIAsuper1300-GFP* vector. The PCR amplification reaction of the downstream cyclization sequence is performed in the same way as the upstream cyclization sequence.

Thirdly, we cloned the *PecircCDPK* looping sequence and its flanking intron, including 171 bp upstream and 137 bp downstream sequences. We introduced *KpnI* and *BsrGI* sites at both ends of the looping sequence along with the flanking intron. The resulting construct was connected to the upstream and downstream cyclization sequence of the *PCAMBIAsuper1300-GFP* vector.

Finally, vector was transformed into *A. thaliana* Columbia-0 with the floral dip method mediated by the *Agrobacterium tumefaciens* strain GV3101. T1 generation transgenic plants were selected on Hygromycin (50 mg/L, Solarbio, China) 1/2 Murashige and Skoog plates (*Clough & Bent, 1998*).

Primers are listed in Table S1.

## Validation of transgenic *Arabidopsis thaliana*

To investigate the role of transgenic plants *PecircCDPK* in regulating the expression of *A. thaliana* related *CDPK* genes, the transgenic T3 seedlings were analyzed using PCR with divergent primers. The homology between *AtCDPK13* and *PH02Gene31251* (the parent gene of *PecircCDPK*) was determined by conducting a BLAST with the *A. thaliana* database (http://www.arabidopsis.org). Subsequently, RT-qPCR was performed on treated and wild-type plants. The qPCR reaction system is: 2× qPCR Master Mix 5.0 µL, forward primer 0.2 µL, reverse primer 0.2 µL, cDNA 0.8 µL, ddH2O 3.8 µL. The qPCR reaction program is: initial denaturation at 95 °C for 30 s (95 °C denaturation for 15 s, 60 °C annealing for 60 s, 60 °C extension for 60 s) × 40 cycles. The relative expression of *AtCDPK13* was calculated using the $2^{-\Delta\Delta Ct}$ method, with *AtACTIN* serving as the reference gene (*Livak & Schmittgen, 2002*; *Shi et al., 2018*). The primers used are listed in Table S1.

## Phenotypic observation and physiology analysis

T3 generation seedlings were initially grown on a 1/2 MS plate for 7 days. Subsequently, they were transferred to a medium with different drought treatments to continue root growth. The root length of both transgenic and wild type plants was observed. The concentration gradient of PEG 6000 used was as follows: 0%, 6%, and 8%. For specific configuration methods, please refer to Verslues (*Verslues et al., 2006*).

Select wild-type and transgenic *Arabidopsis thaliana* at 4 weeks of age to determine leaf water loss rate. The night before the experiment, water the required *Arabidopsis* plants thoroughly and cover them with a film.

Select wild-type and transgenic *Arabidopsis thaliana* plants that have grown for 4 weeks and perform water cut treatment. After 10 days, when the plants show dehydration effects,

collect samples from 3 *Arabidopsis thaliana* seedlings as one sample. Freezing with liquid nitrogen and storing in a −80 °C freezer. We measured the content of malondialdehyde (MDA) and free proline (PRO), as well as the activity of superoxide dismutase (SOD) and peroxidase (POD) (The manufacturer of the commercial reagent kit used was obtained from Nanjing Jiancheng Biotechnology Research Institute in China), and we examined the size of stomata using Environmental SEM.

All statistical analyses were conducted using SPSS software (SPSS, Armonk, NY, USA).

## RESULTS

### Construction of PecircCDPK overexpression vector

The overexpression vector of *PecircCDPK* is made up of by an upstream cyclization sequence, a downstream cyclization sequence and looping sequence with flanking intron (Fig. 1A). The qPCR results indicate that *PecircCDPK* is differentially expressed under drought stress. Then we successfully cloned these sequences (Fig. 1B). Then we used the principle of "using flanking intron sequences supplemented by reverse complementary sequences" to construct the *PecircCDPK* overexpression vector, and the constructed recombinant expression vector was double digested for verification. Electrophoresis showed that there were two bands at about 10,800 bp and 1,900 bp (Fig. 1C). This indicates that the *PecircCDPK* overexpression vector of moso bamboo has been successfully constructed.

### Detection of transgenic *Arabidopsis thaliana* with *PecircCDPK* gene and *AtCDPK13* gene expression

The total RNA of T3 transgenic plants was extracted and then reverse transcribed after treatment with RNase R (*Li et al., 2022*). PCR amplification was carried out using divergent primer primers. The *PecircCDPK* fragment, designed with divergent primers, was successfully amplified from the cDNA of transgenic plant leaves treated with RNase R. However, the *PecircCDPK* fragment with divergent primer design was not observed in the wild-type plants (Fig. 2A). By utilizing the principle of 'flanking intron sequences supplemented by reverse complementary sequences,' the target *PecircCDPK* was introduced into *Arabidopsis thaliana*, resulting in selective splicing and circular formation.

To investigate the potential impact of *PecircCDPK* on gene expression in *Arabidopsis thaliana*, we conducted a BLAST comparison between the parent gene *PH02Gene31251* of *PecircCDPK* and the *Arabidopsis thaliana* database. This analysis identified the gene *AtCDPK13* as having the highest homology with *PH02Gene31251*. Subsequently, a primer was designed for qRT-PCR detection. The results revealed that the expression of the *AtCDPK13* gene was upregulated in transgenic plants compared to wild type plants. Notably, the expression levels of *AtCDPK13* in OE (overexpression)-1 and OE-2 were significantly higher than those in wild type plants. These findings suggest that overexpression of *PecircCDPK* may influence the expression of *AtCDPK13* in *Arabidopsis thaliana*, potentially leading to the regulation of various reactions (Fig. 2B).

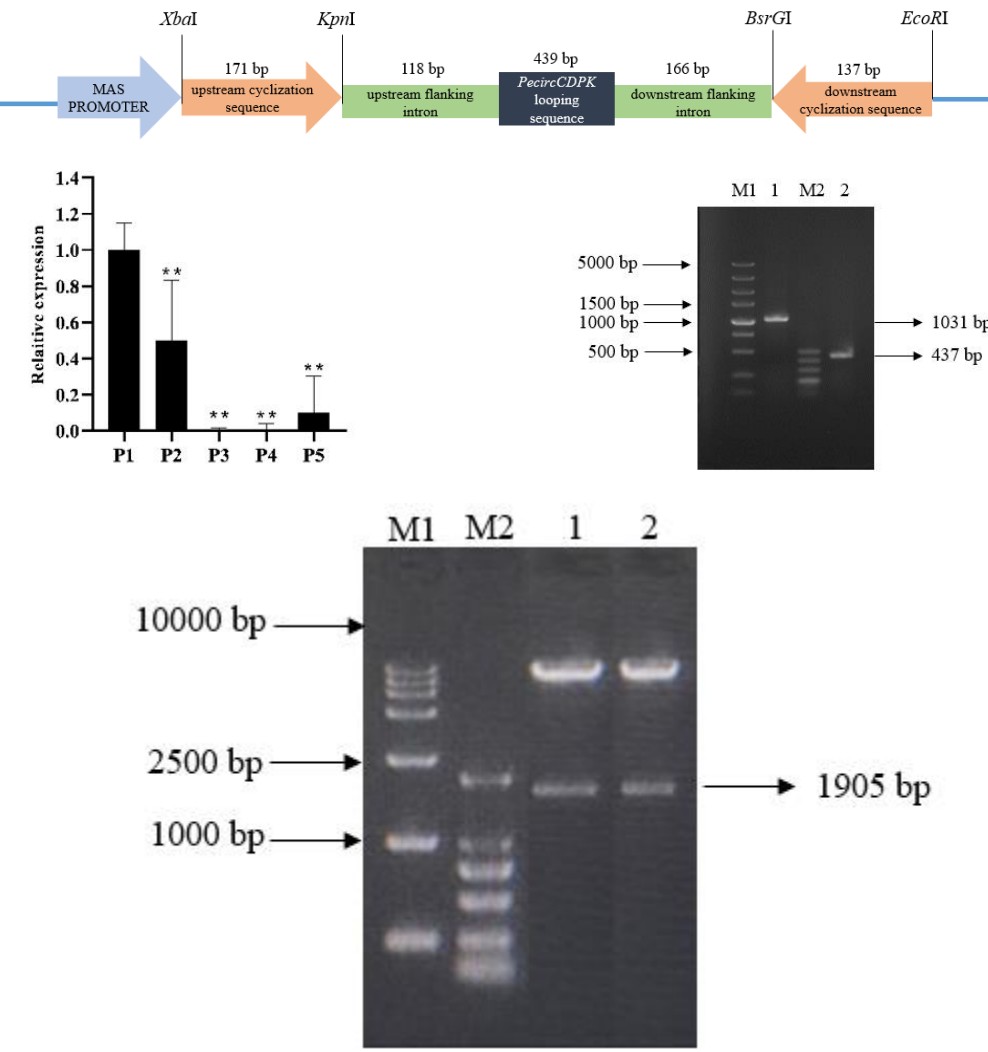

**Figure 1** (A) The overexpression vector of *PecircCDPK*. (B) Relative expression of *PecircCDPK* under the drought stress. Note: ** denotes significant difference at the 0.01 level respectively. (C) Selected intron fragment of moso bamboo *PH02*.

## Response characteristics of transgenic *Arabidopsis thaliana* roots to drought stress

Different concentrations of PEG were added to the culture medium of *PecircCDPK* transgenic and wild-type plants to simulate the effect of drought stress on plant root length (Fig. 3). The growth of wild-type and transgenic plants on the non-stressed 1/2 MS medium was similar, with a relatively small number of roots, indicating that a few long roots were sufficient for water absorption. Under 4% PEG treatment, the total root length of transgenic plants 1 and 3 was larger than that of wild-type plants, while the root length of wild-type plants was significantly shorter. At a high concentration of 8% PEG, the root length of most plants in transgenic line-1 and transgenic line-2 was larger than that of wild-type plants, while the root length of transgenic line-3 plants was larger than that of

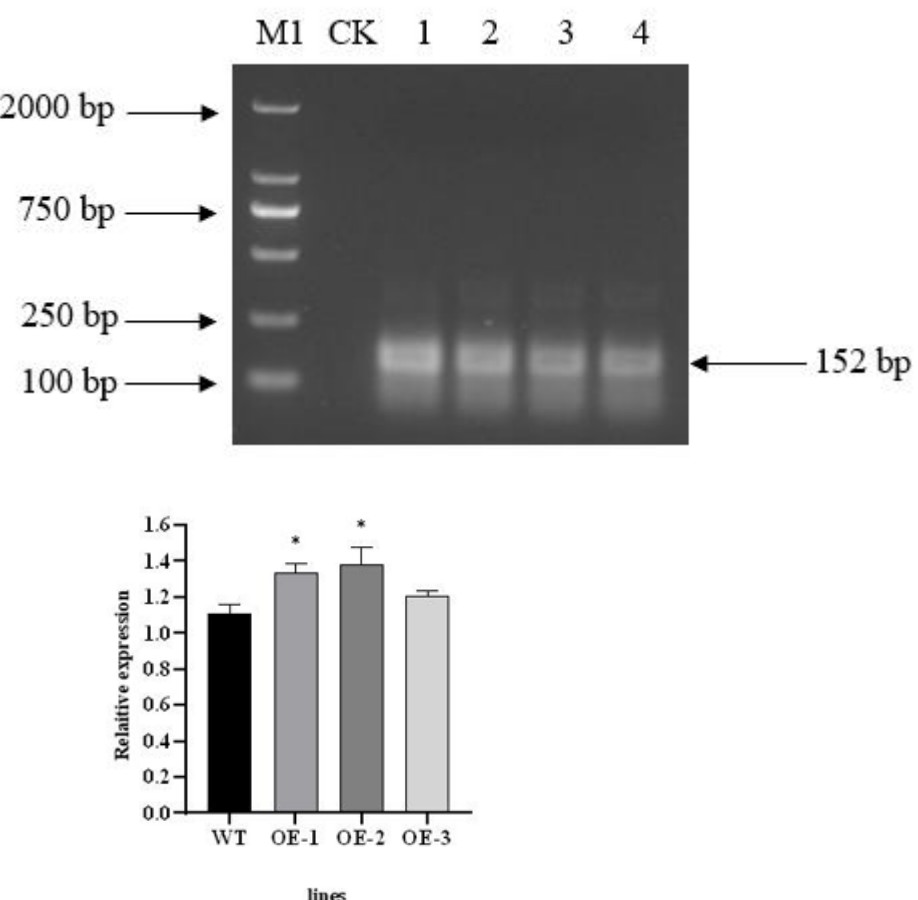

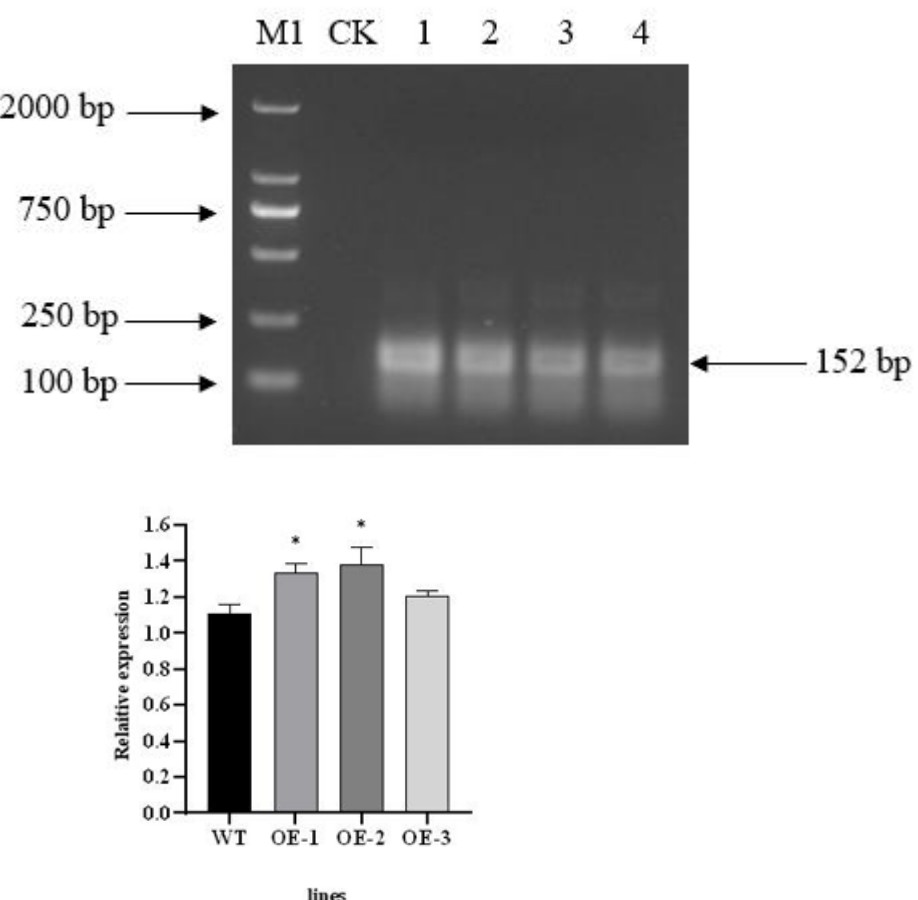

**Figure 2** (A) Using divergent primer to identify transgenic *Arabidopsis thaliana*. Note: M1, DL2000 DNA maker; CK, WT (wild type); 1–4, different transgenic lines. (B) Expression profile of the *AtCDPK13* gene in transgenic *Arabidopsis thaliana*.

wild-type plants, and the root length of wild-type plants decreased. Regarding the number of roots, transgenic plants had more roots than wild-type plants under 6% and 8% PEG treatments. Through comprehensive analysis, it can be concluded that transgenic plants have more developed roots, which allows them to absorb more water under drought stress and improve their resistance to drought.

## Determination of water loss rate and relative water content of transgenic *Arabidopsis thaliana* leaves

Transgenic and wild-type plants that have grown for 4 weeks are selected, weighed, and recorded at different time points. The results are presented in Figs. 4A–4B. Overall, the rate of water loss from the leaves gradually increased over time. In the first 360 min, the water loss rate of transgenic plants at each time point was lower than that of wild-type plants. However, after 360 min, the water loss rate of transgenic plants accelerated and eventually became consistent with that of wild-type plants. By measuring the relative water content of the leaves, it was observed that the water content of transgenic plants increased significantly

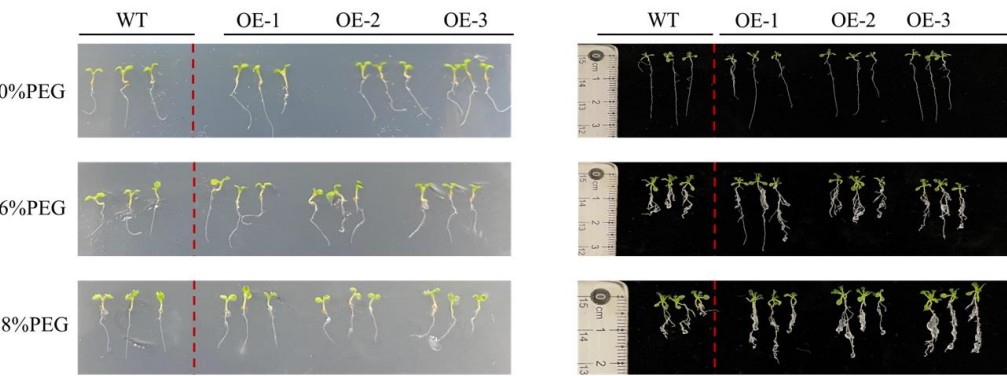

**Figure 3** **Root length characteristics of different *Arabidopsis thaliana* lines under drought stress.** Note: WT (wild type), wild type; OE1-3, different transgenic strains. All materials use three plants as one sample.

and was higher than that of wild-type plants. These findings indicate that the relative water content of transgenic plant leaves has improved. Moreover, the transgenic plants exhibited lower water evaporation compared to wild-type plants in the early stages, suggesting that transgenic plants are better equipped to retain water under adverse conditions, making them more resilient to drought stress.

### Detection of stomata, free proline content and antioxidant enzyme activity in transgenic *Arabidopsis thaliana* under drought stress

Under normal growth conditions, no significant differences were observed in PRO content, MDA content, SOD activity, and POD activity between wild-type plants and transgenic plants. However, after drought treatment, the PRO content in transgenic and wild-type plants showed a significant increase. Specifically, the PRO content in transgenic lines-1 and transgenic lines-2 was significantly higher than that in wild-type plants. The MDA content also increased significantly in both transgenic and wild-type plants, although the increase was smaller in transgenic plants. Notably, the MDA content in wild-type plants was higher than in transgenic plants, and the difference was statistically significant. Both SOD and POD activities showed a significant increase in transgenic and wild-type plants under drought stress conditions, with the POD activity in transgenic plants being significantly higher than that in wild-type plants (Fig. 5). Furthermore, the stomatal size of transgenic *Arabidopsis thaliana* was observed to be smaller after drought stress, which can potentially reduce water evaporation (Fig. 6).

## DISCUSSION
### Efficient and accurate method for constructing circRNA-OE vector

Several plasmid construction methods have been used to cyclize RNA in animal bodies, including complementary sequence mediated cyclization (*Hansen et al.,* ), flanking sequence mediated cyclization (*Kramer et al., 2015*) and circularization of transfer RNA introns (*Noto, Schmidt & Matera, 2017*). Moreover, the *Drosophila* Laccase2 and human zinc finger protein with KRAB and SCANdomains1 flanking intronic sequences have been

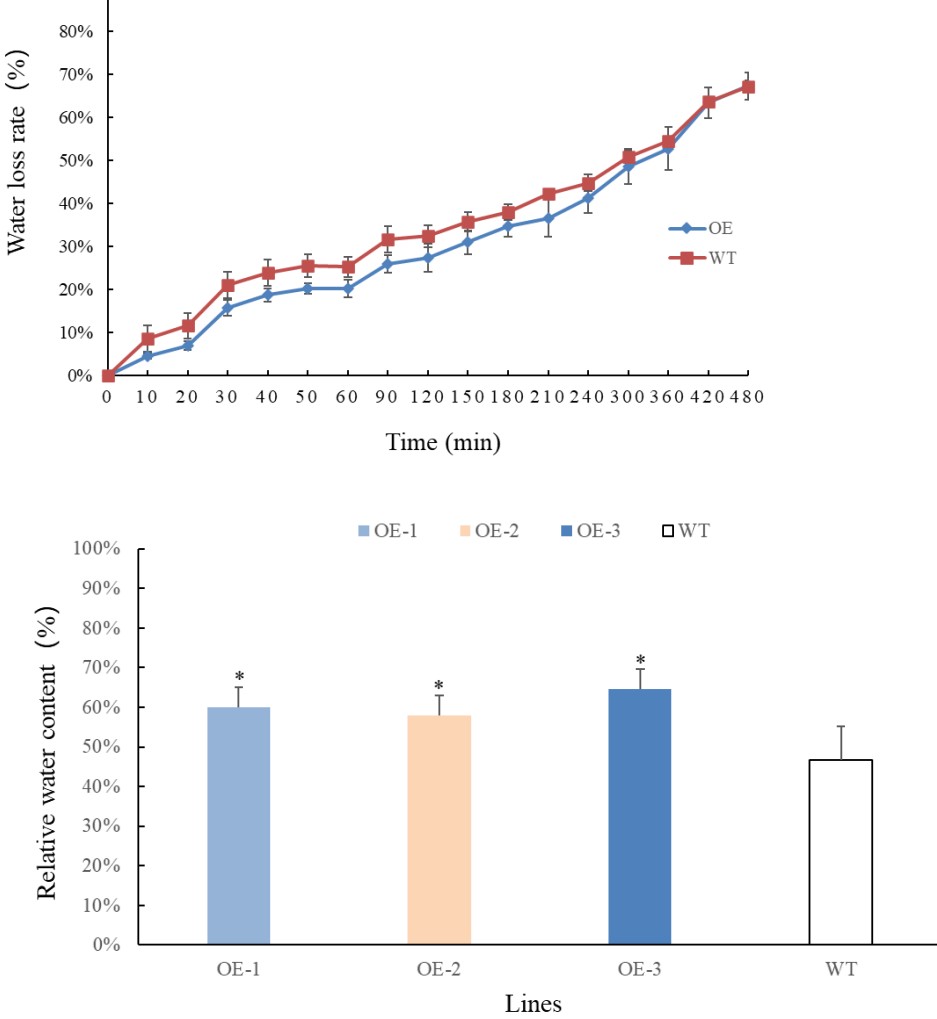

**Figure 4** (A) Comparison of changes in water loss rate in leaves of *PecircCDPK* transgenic *Arabidopsis thaliana*. (B) Comparison of changes in relative water content in leaves of *PecircCDPK* transgenic *Arabidopsis thaliana*.

optimized so that they can efficiently express "designer" exonic circRNAs in human and fly cells (*Kramer et al., 2015*). In addition, an efficient and accurate strategy for overexpressing circRNA in mammals has been developed based on classical complementary sequence mediated cyclization of circRNA (*Liu et al., 2018*). CircRNA has been identified in various plant species, but its function is currently unclear. The lack of this knowledge is partly due to the lack of clear methods for effectively and accurately producing circRNAs in plants. Multiple studies have utilized the feature that paired complementary reverse sequences can promote circRNA biogenesis in animals (*Lu et al., 2015*; *Gao et al., 2019*; *Conn et al., 2017*). However, overexpressing of plasmids can lead to the generation of many undesirable circRNAs in rice (*Lu et al., 2015*), and no Sanger sequencing results concerning

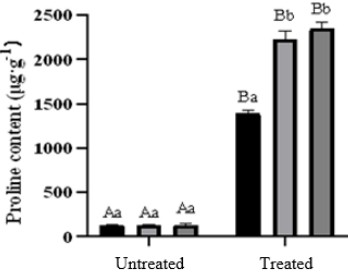
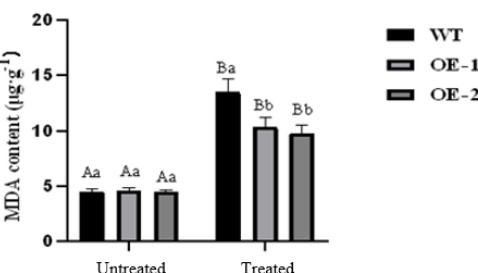
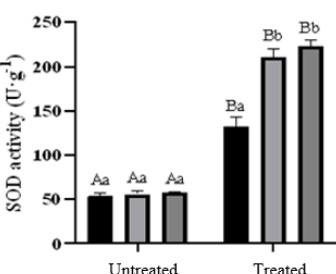
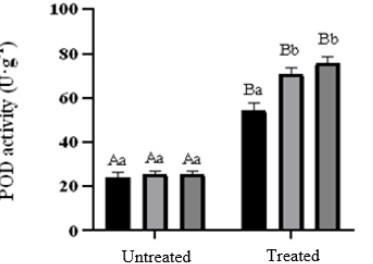

**Figure 5** **Effect of drought stress on proline, MDA contents and activities of SOD, POD in transgenic** *Arabidopsis thaliana.*

back-splicing sites are available for *Arabidopsis* (*Conn et al., 2017*). Therefore, there is an urgent need for an efficient and accurate circRNA expression strategy in plants. Therefore, our study employed the principle of using flanking intron sequences supplemented by reverse complementary sequences to construct the overexpression of circRNA (*Gao et al., 2019*). Finally, divergent primers were used to confirm the transgenic *Arabidopsis thaliana*, indicating successful transfer of the target *PecircCDPK* into *Arabidopsis thaliana*, resulting in a certain degree of selective splicing to form a circular structure.

## Function of *PecircCDPK* in the drought stress response

When exposed to drought stress, plants demonstrate diverse responses. Prior investigations have hypothesized that the parental gene *PH02Gene31251* of *PecircCDPK* functions as a calcium-dependent protein kinase. This protein kinase holds a pivotal position in regulating the root system, stomata, enzyme activity, and other plant responses to drought. These regulatory mechanisms collectively contribute to the enhancement of plants' drought tolerance.

Plants obtain sufficient water through well-developed roots. When faced with water scarcity, plants can adapt to the challenging conditions by adjusting the length and quantity of their roots (*Fang & Xiong, 2015*). In this study, the functional analysis of transgenic plants demonstrated that their root growth ability was enhanced compared to wild-type plants. Measuring the relative water content of plant leaves has been established as a highly effective method for assessing water status (*Torres et al., 2019*). Furthermore, the rate of water loss from plant leaves serves as an indicator of their water-holding capacity, greatly influencing water regulation. Notably, the transgenic plants demonstrated a gradual decrease in the

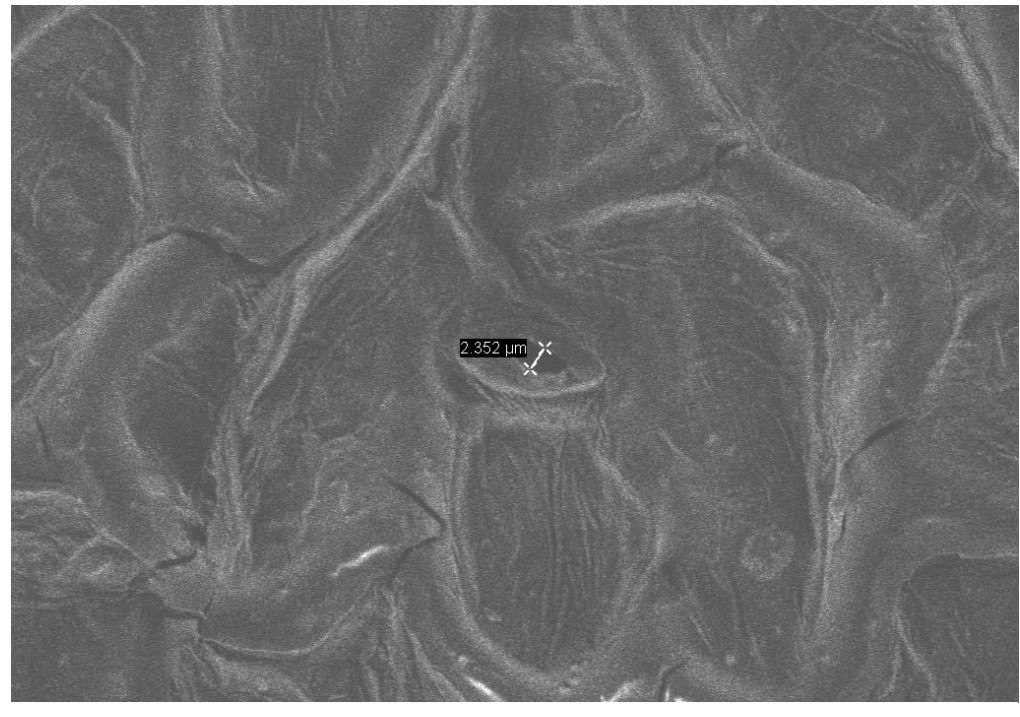

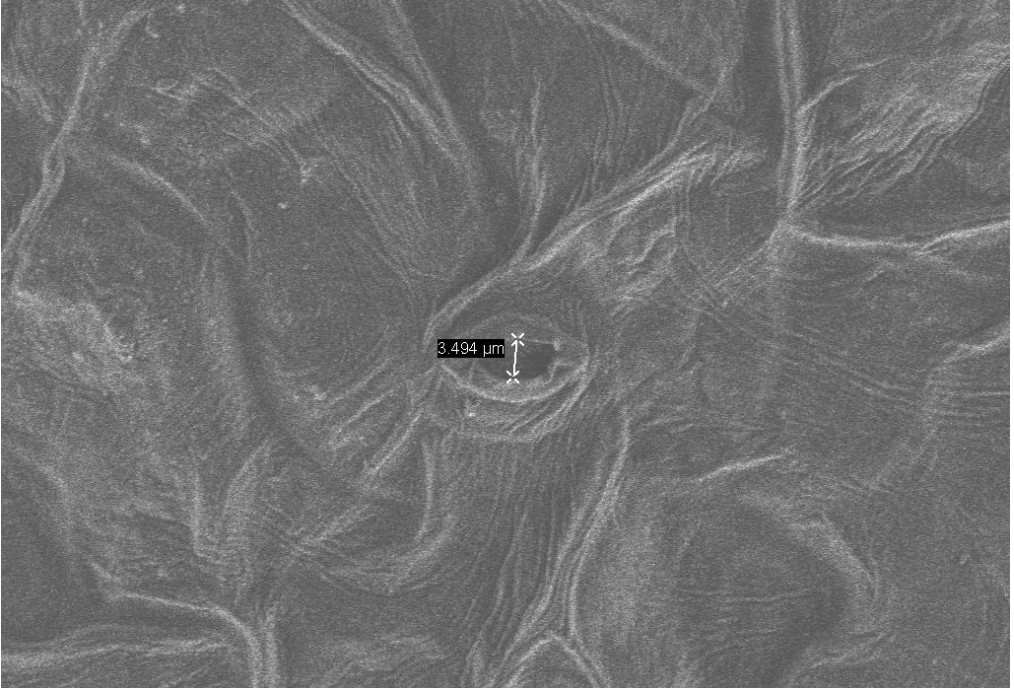

**Figure 6  Environmental scanning electron microscope (ESEM) analysis of *PecircCDPK* transgenic *Arabidopsis thaliana* leaves.**  (A) *PecircCDPK* transgenic *Arabidopsis thaliana*. (B) Wild type.

rate of water loss over time and a notable elevation in the relative water content of their leaves.

When plants are exposed to a water deficit environment, it can cause changes in their metabolic reaction pathways. For instance, the alteration of stoma size and the accumulation and metabolism of proline are closely linked to the mechanisms that plants employ to avoid abiotic stress (*Jiao et al., 2022*). These changes enable plants to retain water within their bodies and reduce water potential, which is crucial for protecting plant cell membranes during drought. *Zhang et al. (2019)* discovered that in transgenic *Arabidopsis thaliana* with excessive expression of the *circGORK* gene, the content of free proline increased 38 times after drought treatment compared to before treatment (*Danan et al., 2011*).

MDA, a commonly used indicator of membrane lipid peroxidation, can effectively indicate the degree of damage to plants under stress conditions (*Wu et al., 2020*). Numerous studies have demonstrated that plants tend to accumulate MDA under drought conditions, and the MDA content is often inversely correlated with plant drought resistance (*Wei et al., 2022*). SOD and POD are key components of the plant antioxidant system and are closely associated with plant drought resistance. Species with strong drought resistance are capable of maintaining high levels of antioxidant enzyme activity for extended periods under drought conditions (*Gharred et al., 2022*). In this study, transgenic lines of *Arabidopsis* exhibited increased proline content, SOD activity, and POD activity under drought conditions. These findings suggest that the heterologous expression of the *PecircCDPK* gene can mitigate biofilm damage to some extent and enhance plant tolerance by increasing the activity of antioxidant enzymes.

However, the importance of this research extends beyond that. Through literature review, it was found that using RNA interference and CRISPR technology, we can also study the role of circRNAs. Additionally, constructing circRNAs deficient mutants can provide further insights into their function. Therefore, in the future, more advanced techniques can be employed to analyze the function of this gene and comprehensively understand the role of *PecircCDPK* in enhancing drought stress resistance in moso bamboo.

## ACKNOWLEDGEMENTS

We would like to express our gratitude to the Key Laboratory of National Forestry and Grassland Administration/Beijing for Bamboo & Rattan Science and Technology for providing experimental equipment.

### Funding

This research was funded by the National Key R&D Program of China (2021YFD2200504_4). The funders had no role in study design, data collection and analysis, decision to publish, or preparation of the manuscript.

### Grant Disclosures

The following grant information was disclosed by the authors:

National Key R&D Program of China: 2021YFD2200504_4.

## Competing Interests

The authors declare there are no competing interests.

## Author Contributions

- Yiqian Li performed the experiments, analyzed the data, prepared figures and/or tables, authored or reviewed drafts of the article, and approved the final draft.
- Wen Xia performed the experiments, analyzed the data, prepared figures and/or tables, and approved the final draft.
- Ying Li conceived and designed the experiments, authored or reviewed drafts of the article, and approved the final draft.
- Xueping Li conceived and designed the experiments, authored or reviewed drafts of the article, and approved the final draft.

## Data Availability

The raw measurements are available in the Supplementary File.

## Supplemental Information

Supplemental information for this article can be found online at http://dx.doi.org/10.7717/peerj.18024#supplemental-information.

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
