# Peer review of "Expression and drought functional analysis of one circRNA PecircCDPK from moso bamboo (Phyllostachys edulis)"

_PeerJ, doi:10.7717/peerj.18024_

## Round 0.1 · original submission · Major Revisions

If you feel you can revise your manuscript according to the reviewers' comments, please revise your manuscript and submit it. Please also send us your written responses to each of the reviewers' comments.

Yours,

Yoshi

Prof. Yoshinori Marunaka, M.D., Ph.D.

Reviewer 1 ·

Basic reporting

The manuscript is well-reported.

Experimental design

Well designed.

Validity of the findings

findings are valid.

Additional comments

My comments
The research work on Expression and functional analysis of one circRNA PecircCDPK from moso bamboo (Phyllostachys edulis) has unveiled the role of circRNA PecircCDPK from moso bamboo. The researchers have conducted a fascinating study that could assist future endeavors of enhancing moso bamboo drought stress tolerance through molecular breeding.
Title: The functional analysis involved only drought stress but not other traits, so I suggest amending the title accordingly.
Keywords should be alphabetical.
Introduction
Lines 40 & 41: You claimed that many studies have investigated the molecular regulatory mechanisms of abiotic stress, but you cited only 1 study, so better to add at least 2 recent related studies.
Line 60 and 61: Is it ideal and good enough studying the functional role of a single circRNA from previously identified 4931 circRNAs?
Line 70: Not clear with; Phenotypic differences in physiological phenotypic were observed.
I believe that many studies have identified and functionally investigated the role of diverse genes that confer drought stress tolerance in moso bamboo when over-expressed. So why do you refrain from mentioning such studies? Like: https://doi.org/10.1007/s00299-020-02625-w.
Materials and methods
Results
Line 205 Detection of stoma? Is it to mean stomata?
Discussion:
Lines 276 and 277 However, the importance of this research extends beyond that. By conducting a literature review and utilizing RNA interference technology, we can investigate the role of circRNAs.
Why not CRISPR than RNA interference Technology? or Are circRNAs not amenable to CRISPR technique?

·

Basic reporting

The present manuscript aims to elucidate the role of PecircCDPK, a circRNA derived from moso bamboo, in response to drought stress. The authors previously predicted that PecircCDPKmight be crucial in plant drought-responsive mechanisms and possess calcium-dependent protein kinase activity. In this study, they cloned PecircCDPKinto a vector, introduced it into Arabidopsis plants, and examined their morphological and biochemical changes. The investigation revealed that expressing PecircCDPK in Arabidopsis enhanced drought tolerance, altered root growth, and affected stomatal behavior and antioxidant enzyme activities. The data generated from this study hold significant potential for future bamboo breeding programs. However, some improvements are desirable.
Abstract

A few points in the abstract could be better organized. For example, the authors stated that PecircCDPK was predicted to be involved in calcium-dependent protein kinase phosphorylation. However, this analysis was conducted previously and not in this study. I suggest moving this information to the second sentence of the abstract to provide background context, including the prediction that PecircCDPK might be involved in drought-responsive mechanisms. This reorganization will help clarify the sequence of research and findings.

The scientific name, i.e., Arabidopsis thaliana, should be italicized. Please check throughout the manuscript.
L148: Please rephrase.
L220: The discussion section could be improved, as some information is repetitive or has already been covered in the Introduction. For example, in Section 4.1, some of this information has been highlighted in the Introduction. Also, the content does not reflect the subheading (efficient and accurate method for constructing circRNA-OE vector).
L230: “Drought stress leads to water loss….” could be separated into a new paragraph.
L264: “MDA, a commonly used indicator…” could be separated into a new paragraph. Also, this should be supported by references. The authors could cite recent publications
L275-281: Perhaps the authors could consider shifting this to a “conclusion” section.
Figure 1: It is recommended to label the relative expression results as “B”, the gel electrophoresis as “C”, and “C” change to “D”. Also, the authors should provide more details in the figure caption. For example, what are P1, P2, P3, P4 and P5?
Figure 3: The images seem small, and I could not see the roots clearly.

Experimental design

Overall, the work has been conducted acceptably, and the methodologies used are relevant to achieving the study objectives. However, some experiments’ descriptions are too brief, and the number of replicates for some experiments is unclear.

The authors stated that moso bamboo seeds were collected from the Guangxi Zhuang Autonomous Region. Does this refer to only one location or different locations in a region? Is it possible to provide GPS coordination?

L81: What were the concentrations for PEG6000?
L83: What do the authors mean by “experimental material”? This sentence could probably be integrated into the following sentence.
L90: “All plant material was...” change to “All plant materials were...”.
L114 and L115: “are” and “is” should be changed to past tense.
L125: Please revise this sentence. Also, what is “reference 10 for more details [15]”?
L135: Please provide the qPCR running conditions.
L146: “obtained from Nanjing Jiancheng Bioengineering Institute”. Does this refer to the manufacturer of the commercial kit used? If yes, please rephrase this sentence for clarity. In contrast, please reference each assay if they were conducted using conventional methods.
L146: The authors should describe how they examined the stomata.  
L146: Please provide a section on “Statistical analyses”, including the statistical software used for statistically significant analysis.

Validity of the findings

L151: The authors should highlight the qPCR results instead of stating they have detected the PecircCDPK expression without further elaboration.
L170: Please provide the value of the highest homology.
L275-281: This paragraph could be the conclusion, but the authors should highlight the major findings first.

·

Basic reporting

The study on circRNA PecircCDPK gene expression from moso bamboo (Phyllostachys edulis) is intriguing, especially with its validation through overexpression studies in Arabidopsis. However, the selection of the candidate gene for this study would have benefited from proper drought stress imposition under water-limited conditions instead of PEG stress, as well as comparative gene expression studies based on previous literature. Additionally, the manuscript requires English editing to enhance clarity and constructiveness. The methods section lacks essential details of protocols, including information on replications and statistical methods. The results section needs improvement to clearly state the study's findings, and the figures should be enhanced for better quality.



The title needs to be changed: If PecircCDPK is one of the circRNA, better mention anyone in the title. It is confusing to read.
Abstract: Missing the background and significance of the drought stress in moso bamboo (Phyllostachys edulis), why they need to be studied, and the connection of circRNAs.
For the first time in the abstract, CircRNA, called PecircCDPK, needs to be abbreviated. What they indicate.
Line 23: Italicize both the Genus and species name for the Arabidopsis thaliana
Introduction:
The introduction lacks the significance of the study and rationale. In the background, there is no information on how drought stress impacts moso bamboo and what the significance is to focus on.
Line 34-35. The sentence is not clear. Please rewrite.

Italicize the botanical names: Also, make sure to italicize both the Genus and species names throughout the manuscripts—ex, Line numbers 56, 129, 130, 132, 165, 167, 168, and 240.

Experimental design

Materials and methods:
The replication information for each experiment is unclear. Words like three pots and one sample are inappropriate for mentioning the replication information. It needs to be specified how many biological and technical replicates were used for each experiment.
Drought stress experiments in moso bamboo are unclear; how was the stress imposed just by adding PEG 6000 to pots or how was it done?
Line 81: Not mentioned what P1 indicates in the treatment.
There is no information available about how the statistical analysis was performed.
Line 92: 2.2 It should be mentioned as RNA extraction and cDNA synthesis instead of RNA and DNA preparation. We are not preparing RNA here.
Line 96. It is unclear here.. DNA examined using NanoDrop2000 means. Is it cDNA?
Line 97-98: Total RNA samples were treated with PrimeSTAR GXL DNA Polymerase (R050A, takara) for reverse transcription. Is that DNA polymerase or reverse transcriptase? Also is this the protocol for cDNA synthesis.

Line 100-102: Hic_scaffold_3:83696771|83697493 was previously validated and predicted to function as a calcium-dependent protein kinase in our previous work [13]. In this study, we have named it PecircCDPK.
Here authors mentioned that they named the Hic_scaffold_3:83696771|83697493 as PecircCDPK. What are the criteria for naming, and what it denotes?
Gene, restriction enzyme, and vector names are italicized in some places, but not in others. Please make sure to be uniform. Ex. PH02Gene31251 and PCAMBIAsuper1300-GFP.
The gene cloning strategy needs to be simplified and written with improved sentences. It is not clear.
Is there any transgene confirmation other than PCR for the expression of PecircCDPK in Arabidopsis?Do authors have confirmation of selectable gene data?
RT-qPCR: It is better to mention the instrument and reagents used to perform the RT-qPCR reaction mixture setup and program details.
Line 125-126: What is the meaning of reference 10 here?
The drought stress impositions and sample collection in Arabidopsis for the physiology experiment are unclear. Is it in the pots? Also, the details of PEG treatments for Arabidopsis transgenics analysis are missing.
The methodology for stomata and microscopic analysis is missing.

Validity of the findings

Results:
3.1 and 3.2: It looks more like methodology; Hence, they should rewrite sentences by emphasizing the results and read better.
The authors mentioned they collected the samples from 4 weeks of seedlings for physiology experiments to measure the water loss and other biochemical assays. They used four-week-old plants to subject the drought stress and collected samples after 10 days, so it's almost 6 weeks, but the seedlings in the pictures are too small. Are there any representative pictures after control and after drought stress impositions to see the morphological differences?
Lines 159-160. What the authors say here is not clear. How can we subject the RNA with RNase treatment will not degrade the RNA? Is that meant to say DNase treatment?
What is the fragment size of the PecircCDPK needs to be mentioned in the results and figure labels.
Line 194: In the results, plants were mentioned to be 3-4 weeks old, and in the methods mentioned 4 weeks old plants. Are they not sure about the experiment samples?
Line 205: Stoma or Stomata. Please correct. The figure did not show the impact of control and drought stress on stomata opening in Arabidopsis WT and transgenics. Was there any difference between control and drought stress conditions? What is the magnification of the stomata microscopic picture?

Figure 1, A: Mention the gene and flanking sequence size in the figure. Also, vector size.
Figure 1, B: In the legends, it needs to be mentioned what the right and left panel figures indicate which data. In the gene expression data, P1-P5 labels, what do they indicate on the x-axis? Specify in the legends.
Figure 1,B, and C: What are the gene product sizes? It should be mentioned on the labels.
Figure 2: what gene product size looks less than 100bp? Why is the gene band degraded on the gel?
Figure 3: What do the right and left panels indicate? In the right panel, no big phenotypic differences were observed between WT and transgenics in 6 and 8% PEG treatments.

Discussion: Missing the correlation between the native gene expression of PecircCDPK with any previously published data.
Missing the connection between PecircCDPK overexpression and discussion of downstream events. Connect the story using previously published data with your results for readers to follow.

Additional comments

Overall, the manuscript needs to be rewritten

---

## Round 0.2 · accepted · Accept

Congratulations again, and thank you for your submission.
Best regards,
Yoshi
Prof. Yoshinori Marunaka , M.D., Ph.D.

·

Basic reporting

The authors have improved and revised the manuscript accordingly. The revised version is much clearer now.

Experimental design

The authors have improved and added statistical analyses for each experiment. The presented methodology is clear.

Validity of the findings

The findings are statistically sound.

Additional comments

There are some very minor errors that the authors probably can make during the proofreading stage. For " example, "made up of by". It should be "made up of". Also, "due to the water consumption". It should be "due to water consumption".